# WHO Grade II or III Solitary Fibrous Tumors (Hemangiopericytomas) of the Spine: Two Case Reports with a Comprehensive Review of the Literature

**DOI:** 10.3390/jcm14176068

**Published:** 2025-08-27

**Authors:** Kazuyuki Segami, Yutaro Okamura, Syu Takahashi, Yasuo Ueda, Koji Kanzaki, Yoshifumi Kudo

**Affiliations:** 1Department of Orthopedic Surgery, Fujigaoka Hospital Showa University, 1-30 Fujigaoka, Aoba-ku, Yokohama City 227-8501, Kanagawa, Japan; y.okamura@med.showa-u.ac.jp (Y.O.); takahashi17615@med.showa-u.ac.jp (S.T.); kkanzaki@med.showa-u.ac.jp (K.K.); 2Department of Pathology, Fujigaoka Hospital Showa University, 1-30 Fujigaoka, Aoba-ku, Yokohama City 227-8501, Kanagawa, Japan; ueda.yasuo.is@med.showa-u.ac.jp; 3Department of Orthopedic Surgery, Showa University, 1-5-8 Hatanodai, Shinagawa-ku 142-8555, Tokyo, Japan; kudo_4423@med.showa-u.ac.jp

**Keywords:** solitary fibrous tumor, hemangiopericytoma, spine

## Abstract

Solitary fibrous tumors (SFTs) of the spine are rare. SFTs, especially those classified as WHO grade II or III (previously termed hemangiopericytomas), are aggressive neoplasms with a high recurrence rate and metastatic potential. In the literature, descriptions of SFTs are limited to case reports and small case series. To our knowledge, 157 cases, including the current case, have been reported since Schirger’s 1958 report on spinal SFTs. This report describes two cases of WHO grade II and III SFTs in the spine and presents a review of the literature. In the first case, an extradural WHO grade II SFT recurred 6 years after the first surgery, and a second surgery was performed, including wide excision of the surrounding tissue. The patient has remained recurrence-free for 16 years since the second surgery. In the second case, an intradural extramedullary WHO grade III SFT was resected, including the dura mater, and the patient has remained recurrence-free for 3 years since the surgery. Few reports have described tumor recurrence and long-term outcomes after reoperation, as in the first case, or extensive resection including the dura, as in the second case. Furthermore, the literature review not only summarizes patients’ general and surgical information, but also indicates, based on multivariate analysis, that gross total resection (GTR) is an important factor in preventing recurrence and metastasis. This is the first study to comprehensively examine previous reports and identify risk factors for recurrence and metastasis. In addition, because recurrences have been reported long after surgery, we believe that even if GTR is performed surgically, it is important to conduct follow-ups to check for long-term recurrence.

## 1. Introduction

Solitary fibrous tumors (SFTs) and hemangiopericytomas (HPCs) were previously considered different types of tumors. As both tumors share inversions at 12q13 leading to STAT6 nuclear expression, the World Health Organization Classification of Tumors of the Central Nervous System (CNS WHO) assigned the combined term SFTs/HPCs in 2016 [1]. The term “hemangiopericytoma” has been revoked in the 2021 WHO Classification of CNS Tumors, with the tumor now being considered an SFT [2]. To create a single designation for tumors in the spectrum of low-grade SFTs and higher-grade lesions previously designated as HPCs and anaplastic HPCs, the CNS WHO assigns three grades for SFTs: grade I—highly collagenous, relatively low cellularity, spindle cell lesion previously diagnosed as an SFT; grade II—more cellular, less collagenous tumor with plump cells and “staghorn” vasculature that was previously diagnosed as an HPC; and grade III— termed an anaplastic HPC in the past, diagnosed on the basis of >5 mitoses per 10 high-power fields [1]. Although these tumors are understood to share a biologic basis, they remain behaviorally distinct; WHO grade I SFTs occurring in the CNS tend to be classified as benign [3], while WHO grade II or II are aggressive, known to metastasize systemically, and commonly recur [4,5,6].

SFTs are rare CNS tumors [7] that are commonly located in the brain, with occurrence in the spine being extremely rare [8]. This case report presents two targeted cases of WHO grade II and III SFTs in the spine (an extradural tumor with recurrence 6 years later and an intradural extramedullary tumor) and reviews the literature on the management of WHO grade II or III spinal SFTs. There are few reports of extradural tumor recurrence and long-term outcomes after reoperation, and intradural extramedullary tumors that have been resected, including the dura mater. Furthermore, the literature review summarizes patient demographics, pathology, and surgical outcomes, exploring factors contributing to recurrence. This is the first study to comprehensively examine previous reports and describe risk factors for recurrence and metastasis.

## 2. Case Reports

### 2.1. Case 1

*History.* A 40-year-old woman with a history of left leg pain underwent a diagnostic workup, including thoracolumbar computed tomography (CT) myelography, which revealed an intraspinal extradural tumor at T12–L1 (Figure 1). MRI was not performed, as this evaluation occurred 22 years ago.

*Operation and Pathological Examination.* The tumor was resected. Following T12 and L1 laminectomies, a brownish-red tumor covered by a thin capsule was observed on the dural tube. The tumor was well circumscribed, had a smooth margin with dural attachment, and was scarcely vascularized. The tumor did not adhere to the surrounding structures, including the T12 nerve root. Microscopic examination revealed tumor cell proliferation around the branched vessels. Oval- or spindle-shaped cells with a slightly acidic cytoplasm and oval nuclei were observed. No necrosis was observed. However, scattered mitotic figures were observed. Immunohistochemical staining was positive for STAT 6 and negative for CD 34. The Ki-67 labeling index was 2–3%. A pathological diagnosis of a WHO grade II SFT was considered (Figure 2).

*Postoperative Course.* The patient was treated with teceleukin-based chemotherapy. Pain in the left leg recurred 6 years after the first surgery. Neurological deficits were not observed; however, magnetic resonance imaging (MRI) revealed tumor recurrence (Figure 3). Considering the aggressiveness, high recurrence rate, and need for a second surgery, gross total resection (GTR) including the neighboring structures, such as the T12 and L1 vertebrae, laminae, dura mater, and right T12 root, was performed (Figure 4). The pathological findings included WHO grade II SFT recurrence. No tumor recurrence or metastasis was observed during the 16-year follow-up period after the second surgery.

### 2.2. Case 2

*History.* A 36-year-old woman reported back pain, numbness, and muscle weakness in both legs that developed gradually. MRI revealed a tumor with iso-signal intensity on both T1- and T2-weighted imaging posterior to the thecal sac at T3–4. Gadolinium-enhanced MRI revealed uniform enhancement of the tumor and a dural tail sign, which was highly suggestive of a meningioma (Figure 5).

*Operation and Pathological Examination.* The tumor was resected. T3 and T4 laminectomies were performed to access the spinal canals. Dura mater incision revealed an oval, brownish-red tumor covered by a thin capsule in the subarachnoid space. Because the tumor strongly adhered to the right portion of the dura mater, we first resected the T3/4 right facet joint to secure an operative space, and the tumor and neighboring dura mater were subsequently resected en bloc. The dural defect was covered with artificial dura mater, and the resected T3/4 facet joint was reconstructed using a pedicle screw (Figure 6). Microscopic examination revealed tumor cell proliferation around the branched vessels. The sinus-like vessels demonstrated a “staghorn” pattern typical of HPCs. The tumor cells were spindle-shaped with oval nuclei and mitotic figures (>5 mitoses per 10 high-power fields). Immunohistochemical staining was positive for STAT 6 and negative for CD 34. The Ki67 labeling index was 6–7%. Thus, a pathological diagnosis of a WHO grade III SFT was considered (Figure 7).

*Postoperative Course.* The patient’s neurological symptoms resolved postoperatively. No tumor recurrence or metastasis was observed during the 3-year follow-up period.

## 3. Review of the Literature

### 3.1. Literature Search

PubMed, Scopus, and Embase searches were performed for HPCs, HPCs/SFTs WHO grade II or III, and WHO grade II or III SFTs of the spine (Appendix A). Additionally, the reference sections of each study were reviewed, to identify additional cases. The inclusion criteria were that the primary tumor was in the spine, and that reports included not only surgical cases but also cases treated with radiotherapy (RT), chemotherapy, and no treatment. Exclusion criteria included reports of distant metastasis to the spine from other sites and recurrence after tumor removal (those without detailed descriptions of the initial treatment). We identified 62 previously published papers with 170 cases [4,6,9,10,11,12,13,14,15,16,17,18,19,20,21,22,23,24,25,26,27,28,29,30,31,32,33,34,35,36,37,38,39,40,41,42,43,44,45,46,47,48,49,50,51,52,53,54,55,56,57,58,59,60,61,62,63,64,65,66,67,68]. Fifteen reported cases [11,15,31,38,43,44,47,48,50,54,55,68] that focused on local recurrence or metastases to the spine were eliminated. A total of 157 reported cases of WHO grade II or III spinal SFTs were found, including our report and those previously diagnosed as spinal HPCs and WHO grade II or III HPCs/SFTs (Table 1).

### 3.2. Summary of General and Surgical Information of Patients

#### 3.2.1. Age and Sex

One hundred and twenty-eight patients reported their age at presentation. Most cases occurred in adults (85.2% of patients aged > 21 years). The patients’ ages ranged from 2 to 82 years (median: 40.8 years). One hundred and twenty-eight cases included information on the patient’s sex. A slight male predominance was observed among the reported cases (43.3% male vs. 38.2% female; Table 2).

#### 3.2.2. Spinal Level and Compartment

All the cases included information on the spinal levels, except for two cases; the plurality of lesions occurred in the thoracic (40.1%), cervical (24.8%), and lumbar (15.3%) regions. The remaining tumors occurred in the sacrum or across the junctional levels. A total of 113 reported cases contained information on the spinal compartments involved. Based on the morphological data in the literature [6,9,10,11,12,13,14,15,16,17,18,19,20,21,22,23,24,25,26,27,28,29,30,32,33,34,35,36,37,39,40,41,42,45,46,47,49,50,51,52,53,54,56,57,58,59,60,61,62,63,64,65,66,67], spinal SFTs were divided into six types (Figure 8): Type 1, intracanal–intradural–extramedullary tumors (Case 2); Type 2, intracanal–intradural–intramedullary invasion tumors; Type 3, intracanal–extradural tumors (Case 1); Type 4, extracanal (vertebra, spinous process to lamina, intraforaminal, intraforaminal to paravertebra) tumors; Type 5, extradural to extracanal tumors; and Type 6, intradural to extracanal tumors. The plurality was Type 1 (21.7%) and Type 5 (21.0%), followed by Type 2 (8.9%) and Type 6 (7.7%).

#### 3.2.3. MRI Appearance and Differential Diagnosis

Ninety-three reported cases contained information on the MRI appearance. In all the cases, the tumors exhibited marked enhancement with intravenous contrast. Most lesions were hypointense or isointense on the T1-weighted images, and either isointense or hyperintense on the T2-weighted images. Some reports mentioned the preoperative diagnoses, which were most often meningioma or schwannoma [29,32,33,37,39,41,46,49,51,58,59,60,66,67]. One large sacral tumor was presumed to be either a chordoma or a giant cell tumor [39]. In cases of irregular bone invasion, Jia et al. [54] reported the radiological differential diagnoses to include giant cell tumors, malignant schwannomas, benign hemangiomas, and other metastatic tumors.

#### 3.2.4. Treatment

A hundred and fifty-five of the reported cases contained information regarding the treatment. All the patients underwent surgery, except for three patients, two of whom received only radiation [12,40] and one of whom received radiation and chemotherapy [40]. One hundred and forty-seven reported cases contained detailed information on the surgery. GTR was performed in 58.0% of the patients, and subtotal resection (STR) or partial resection (PR) was performed in 35.7% (Table 3). A total of 14 patients (12 GTR, 1 STR, and 1 PR) underwent preoperative embolization of the tumor [18,25,30,36,47,49,54]. A total of 70 patients (36 GTR, 28 STR, 5 PR, and 1 surgery) underwent combined radiation [9,10,15,16,18,20,22,23,24,25,27,28,32,34,36,39,46,47,50,54,59,60,62,63], and 4 patients (3 GTR and 1 STR) underwent combined radiosurgery [46]. A total of 5 patients (3 GTR and 1 STR), including the patient in Case 1, underwent combination chemotherapy [23,46,50,63].

#### 3.2.5. Tumor Origin

Few reports have indicated intraoperative assessments of the tumor origin [9,11,29,34,47,65]. Fifteen patients had tumors arising from the vertebral body [34], three from the nerve roots [9,29,65], and two from the spinal cord [11,47]. Fourteen patients, including the patient in Case 2, had tumors adherent to the dura [11,12,15,34,49,50], and five patients had tumors involving the nerve roots [20,42,57,62,64].

#### 3.2.6. Histopathology

All cases indicated tumor histopathology except for one. The WHO grade was known for 76 patients. Thirty-nine patients (24.8%) had WHO grade II tumors, thirty-five patients (22.3%) had WHO grade III tumors, and two patients had borderline tumors between grades II and III (Table 3).

#### 3.2.7. Clinical Outcome

One hundred and eight patients provided clinical outcome data. The follow-up period ranged from 1 month to 31 years. Seventeen patients had undergone a follow-up of <1 year, and eighty-one patients had undergone a follow-up of <5 years. Forty-six patients (29.3%), including the patient in Case 1, exhibited tumor recurrence or progression [9,10,18,19,20,23,30,34,40,46,50,53,54,63,66], and nine patients (5.7%) had metastasis [18,40,46,47,54]. Generally, outcomes depended largely on the histology and extent of resection. Thirty-three patients with WHO grade III tumors had reported the clinical outcome data. Nineteen patients (57.6%) had local recurrence [46,50,54,66], and four patients (12.1%) had distant metastasis [46,47] (including three patients who had both local recurrence and metastasis [46]). Thirty-nine patients with WHO grade II tumors had reported the clinical outcome data. Sixteen patients (41.0%) had local recurrence [46,53,54,66], with no distant metastasis. Furthermore, the outcomes seemed to depend largely on the extent of resection. Clinical outcome information was available for 14 patients with WHO grade III treated with STR or PR. Twelve patients (85.7%) had tumor recurrence or progression [46,50,54,66], and one of them had both local recurrence and metastasis [46]. Outcome information was available for 19 patients with WHO grade III treated with GTR. Seven patients (36.8%) exhibited tumor recurrence or progression [46,54], and three patients had metastasis [46,47] (including two patients who had both local recurrence and metastasis [46]). Outcome information was available for 11 patients with WHO grade II treated with STR or PR. Seven patients (63.6%) had tumor recurrence or progression [46,54,66], with no distant metastasis. The clinical outcome information was available for 28 patients with WHO grade II treated with GTR. Nine patients (32.1%), including the patient in our case report, had tumor recurrence or progression [46,53,63], with no distant metastasis.

### 3.3. Comparison Between Recurrence or Metastasis Cases and Disease-Free Cases

A statistical analysis was performed to determine the factors that influenced the outcome of either tumor recurrence or metastasis. Significance was set at *p* < 0.05. Statistical analyses were conducted using JMP Pro 17 Software (SAS Inc., Cary, NC, USA) for Windows. Table 3 shows the results of univariate analysis. The differences between the recurrence or metastasis cases and disease-free cases were analyzed using Pearson’s Chi-square test for nominal variables. Only the extent of resection was significantly associated with recurrence or metastasis (*p* < 0.0001). To identify the most significant risks for recurrence or metastasis, risk factor analysis was performed using multivariate logistic regression analysis (Table 4). A significant difference was noted in the extent of resection.

**Table 3 jcm-14-06068-t003:** Differences between recurrence or metastasis and disease-free cases.

		Outcome								
Factors		All		Recurrence or Metastasis		Disease-Free		Unknown		
		Count	%	Count	%	Count	%	Count	%	*p*-Value
Age		40.8 (yrs)		39.6 (yrs)		41.3 (yrs)				0.61
Sex	Male	68	43.3	27	39.7	31	45.6	10	14.7	0.42
	Female	60	38.2	19	31.7	30	50.0	11	18.3	
	Unknown	29	18.5	0	0.0	0	0.0	29	100.0	
Location	Occipitocervical or cervical	43	27.4	16	37.2	12	27.9	15	34.8	0.19
	Cervicothoracic or thoracic	66	42.0	15	22.7	32	48.5	19	28.8	
	Thoracolumbar or lumbar	32	20.4	12	37.5	12	37.5	8	25.0	
	Lumbosacral or sacral	14	8.9	2	14.3	4	28.6	8	57.1	
	Unknown	2	1.3	1	50.0	1	50.0	0	0.0	
Classification	Types 1–3 (intracanal)	57	36.3	20	35.1	31	54.4	6	10.5	0.38
	Types 4–6 (expanding widely)	56	35.7	23	41.1	25	44.6	8	14.3	
	Unknown	44	28.0	3	6.8	5	11.4	36	81.8	
Extent of resection	GTR	91	58.0	20	22.0	48	52.7	23	25.3	<0.0001
	STR or PR	56	35.7	24	42.9	11	19.6	21	37.5	
	Others	10	6.3	2	20.0	2	20.0	6	60.0	
WHO grade	Grade II	39	24.8	16	41.0	23	59.0	0	0.0	0.0967
	Grade III	35	22.3	20	57.2	13	37.1	2	5.7	
	Borderline between grade II and III	2	1.3	1	50.0	1	50.0	0	0.0	
	Unknown	81	51.6	10	12.1	25	30.1	48	57.8	
Adjuvant radiotherapy (GTR)	GTR	47	60.3	8	17.0	27	57.5	12	25.5	0.3
	GTR + RT	31	39.7	10	32.2	19	61.3	2	6.5	
Adjuvant radiotherapy (STR or PR)	STR or PR	25	45.5	7	28.0	6	24.0	12	48.0	0.21
	STR or PR + RT	30	54.5	15	50.0	5	16.7	10	33.3	

**Table 4 jcm-14-06068-t004:** Multiple logistic regression analysis of recurrence or metastasis.

Factors	Odds Ratio	95% Confidence Interval	*p*-Value
Age	0.99	0.95–1.03	0.57
Sex (male)	1.46	0.48–4.49	0.5
Location			
Occipitocervical or cervical	3.39	0.89–12.90	0.07
Cervicothoracic or thoracic	Reference		
Thoracolumbar or lumbar	1.8	0.40–8.11	0.44
Lumbosacral or sacral	0.53	0.02–16.40	0.71
Classification (types 4–6)	1.54	0.49–4.84	0.46
Extent of resection (STR or PR)	6.35	1.77–22.77	0.0045
Adjuvant radiotherapy (without radiotherapy)	0.68	0.22–2.17	0.52
WHO grade III	1.69	0.53–5.50	0.38

## 4. Discussion

SFTs are mesenchymal tumors that were first described in 1931 by Klemperer and Coleman as primary neoplasms arising from the pleura [69]. HPCs, which are derived from the pericytes surrounding the capillaries and post-capillary venules, were first described by Stout and Murray in 1942 [70]. Although both SFTs and HPCs can arise in any soft tissue of the human body, the CNS WHO assigned the term SFTs to such lesions in 2021, as both the tumors share inversions at 12q13 leading to STAT6 nuclear expression. WHO grade II or III SFTs are well-recognized aggressive neoplasms with a high rate of recurrence and propensity to metastasize [6].

The majority of spinal WHO grade II or III SFT cases have occurred in adults, with a slight male predominance. These tumors most often occur in the thoracic region but can also occur in a wide variety of locations, from the intradural to the vertebral regions. 

SFTs did not show any specific imaging features. Intracanal tumors, such as Types 1, 2, and 3, were sometimes misdiagnosed preoperatively as benign tumors, such as schwannomas or meningiomas, owing to non-involvement of the surrounding bone. Residual SFTs from incomplete tumor removal can spread to the spinal cord and other tissues, causing tumor recurrence and regrowth [46,50]. Thus, distinguishing spinal SFTs from other types of tumors preoperatively is essential for improving clinical outcomes. Okubo et al. [66] reported that most patients with SFTs demonstrated isointensity in the spinal cord on both T1- and T2-weighted images compared to schwannomas, and no significant differences compared to meningiomas. Nearly all the patients with SFTs demonstrated highly uniform enhancement patterns, similar to those with meningiomas. Compared to meningiomas, SFTs lack the dural tail sign and intratumoral calcification, and have an irregular shape. However, the dural tail sign can appear when the SFTs arise in the dura mater [64], as in Case 2. The tumor often demonstrates marked enhancement with prominent flow voids on MRI owing to its hypervascularity [36]. However, cases that do not show such imaging features have been reported, which often mimic various tumors such as meningiomas and schwannomas on imaging [46,47].

Total resection is the gold standard of treatment for SFTs. Some studies demonstrated better recurrence-free and overall survival rates following GTR of SFTs in the brain [71,72]. Schiariti et al. [71] reported that patients who underwent GTR demonstrated longer mean overall survival (GTR 235 vs. STR 175 months; *p* = 0.47) and mean recurrence-free intervals (GTR 117 vs. STR 53 months; *p* = 0.0045). Soyuer et al. [72] reported 5-year local control rates in patients treated with GTR and STR of 84 and 38%, respectively (*p* = 0.0034); few studies found similar findings in spinal SFTs [50,54,73]. However, other studies did not find any clinical, overall, or recurrence-free survival benefits following complete resection of spinal SFTs [46,59,74]. Liu et al. [46] reported that total resection in high-risk patients may not be necessary for spinal SFTs, especially those that are difficult to completely remove, such as tumors that had grown out from the spinal canal, because total resection did not represent a significant survival or recurrence-free benefit. They described stereotactic radiosurgery for residual tumors that had a satisfactory tumor control effect. Li et al. [59] also proposed that subsequent adjuvant RT is more beneficial than total resection for patients at high risk of injury to the nerve root or dura mater. As in Case 2, when the tumor is tightly adhered to the dura mater in a location where it is relatively easy to remove, such as within the spinal canal, the possibility that the dura mater is the tumor origin cannot be ruled out, so removal of the dura mater as well should be considered.

RT for SFTs in the brain improves local control and overall survival [4,75,76]. However, whether postoperative RT should be performed for spinal SFTs remains unknown. Studies demonstrated no significant difference in the outcomes of patients with spinal SFTs who received adjuvant radiotherapy when compared to those of patients who did not receive RT [37,46,74], and this is consistent with our findings. Using precision radiotherapy, such as radiosurgery with high radiation doses, the morbidity associated with adjacent organ damage, especially spinal cord toxicity, is minimized [77]. Combs et al. [77] analyzed the use of high-precision RT in patients with SFTs, including two spinal cases. They demonstrated overall survival rates of 100% and 64% at 5 and 10 years, respectively, in patients treated with a combination of surgical and RT approaches. Furthermore, for recurrent SFTs of the CNS including spine cases undergoing radiosurgery, tumor control rates between 82% and 93% have been reported [74,78,79]. Ecker et al. [74] suggested radiosurgery to be a good alternative to repeated surgery in patients with recurrent SFTs. However, adjuvant chemotherapy has generally been proven to be ineffective [46,50,54,74].

The 5-year recurrence-free rate of intracranial SFTs was 46–89% in recent reports [71,74]. Liu et al. [46] reported 24 patients with intraspinal SFTs and demonstrated a 5-year recurrence-free survival rate of 29.4%, which is much lower than that of intracranial SFTs. Intraspinal SFTs can recur more often or earlier than intracranial SFTs. Several studies have shown that a higher pathological grade is associated with a greater local recurrence rate in intraspinal SFTs [37,46,54,73]. In contrast, our analysis did not identify pathological grade as a significant risk factor for recurrence or metastasis. One reason for this apparent discrepancy may be that pathological classification was not recorded in 81 cases (51.6%). However, consistent with our findings, total resection appears to significantly decrease the recurrence rate of spinal SFTs [54,73]. High-grade intraspinal SFTs may exhibit firm adherence to the dura mater, irregular invasion of the bone, and rich vascularization; thus, a small operating field in the spine may make complete removal of the tumor challenging. Recurrence often occurs many years after initial treatment [9,11], necessitating careful long-term follow-up of all patients with SFTs. In this review, distant metastases of spinal SFTs following initial treatment were observed in only nine cases (5.7%), a rate lower than that reported for intracranial SFTs, with distant metastases ranging from 14% to 50% [4,42,74]. Reports of survival rates for intraspinal SFTs are rare, but two available studies indicated a 76% 5-year survival rate [46] and an 85% 10-year survival rate [8].

Comparisons between intracranial and intraspinal SFTs are also uncommon. Boyett et al. [8] reported that intraspinal SFTs had a smaller tumor size, lower malignancy, and a lesser extent of resection compared to intracranial SFTs. They also found an association between spinal tumor location and improved survival in patients with SFTs of the CNS, consistent with the results from Giordan et al. [73]. As mentioned above, spinal tumors have a superior survival rate despite being more likely to recur. Local recurrence is expected if GTR cannot be performed on spinal tumors, but the malignancy is low. Hence, even such cases may not lead to fatal problems such as distant metastasis.

One of the limitations of this study is that the factors used in the multivariate analysis, particularly the WHO grades, contained a large number of missing data, which may have influenced the results. By continuing to increase the number of cases, missing data will be supplemented, and future research will involve creating treatment algorithms based on tumor location and malignancy.

## 5. Conclusions

Spinal SFTs of the spine are rare neoplasms. To date, no large case series has been performed. Spinal SFTs are less malignant than cranial SFTs, and patients have higher rates of long-term survival. However, spinal SFTs, especially WHO grade II or III, are aggressive neoplasms with a high recurrence rate. GTR is recommended to treat these lesions. When GTR is difficult, such as in recurrent cases, radiosurgery is one of the most important alternatives for treating spinal SFTs. Again, the reported incidence of local recurrence for this type of tumor is very high, and recurrence often occurs many years following the initial treatment. Therefore, careful long-term observation is necessary following surgery even if the tumor has been completely resected.

## Figures and Tables

**Figure 1 jcm-14-06068-f001:**
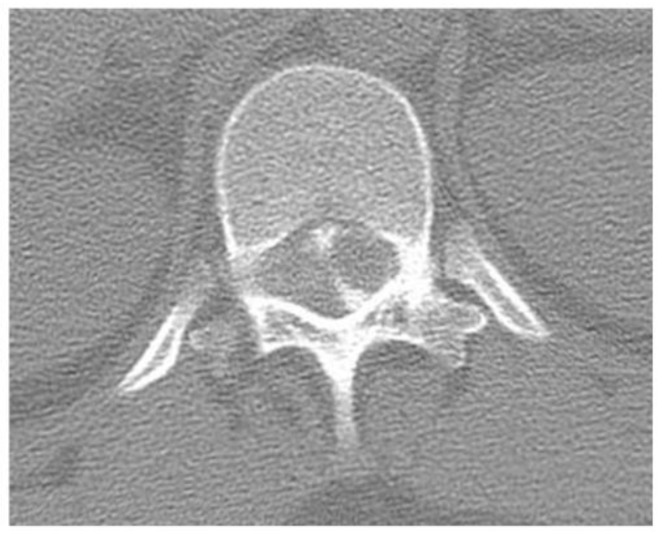
Preoperative computed tomography myelography demonstrating an intraspinal extradural tumor at T12–L1.

**Figure 2 jcm-14-06068-f002:**
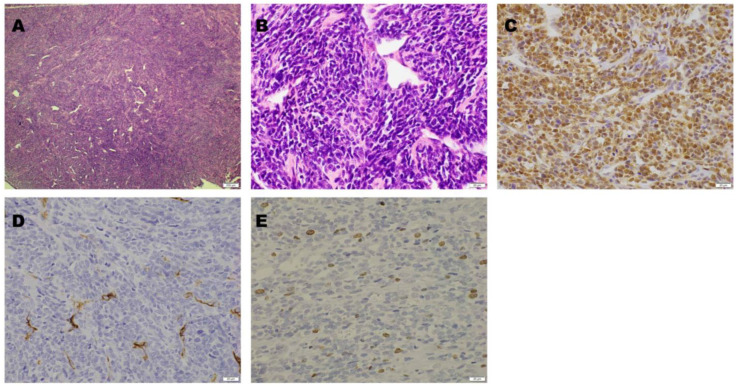
Microscopic findings. Hematoxylin and eosin staining revealed tumor cell proliferation around the branched vessels ((**A**): magnification, 40×). The tumor cells were oval- or spindle-shaped with slightly acidic cytoplasm and oval nuclei ((**B**): magnification, 400×). On immunohistochemical staining, STAT6 was positive ((**C**): magnification, 400×), while CD34 was negative ((**D**): magnification, 400×). The Ki-67 labeling index was 2–3% ((**E**): magnification, 400×).

**Figure 3 jcm-14-06068-f003:**
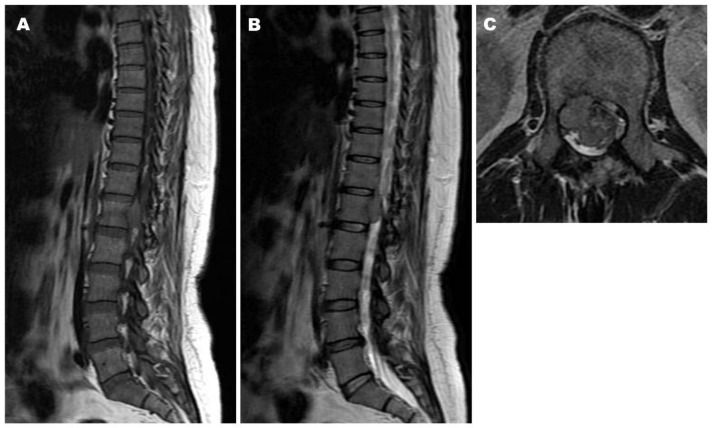
Magnetic resonance imaging (MRI) six years after the first surgery demonstrating tumor recurrence at T12–L1, the same level as before. Iso intensity on both T1- and T2-weighted images ((**A**): Sagittal T1-weighted image, (**B**): Sagittal T2-weighted image, (**C**): Axial T2-weighted image).

**Figure 4 jcm-14-06068-f004:**
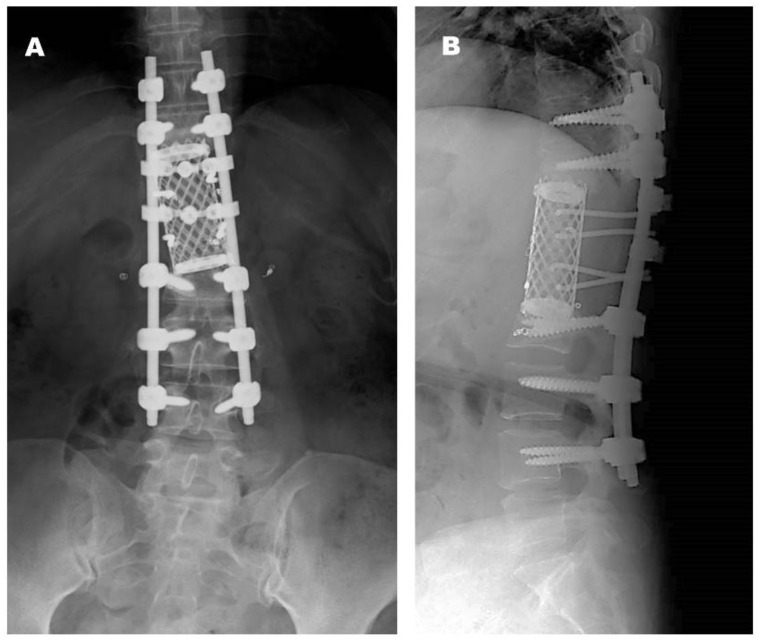
Radiographs obtained at final follow-up after the second operation ((**A**): anteroposterior view, (**B**): lateral view).

**Figure 5 jcm-14-06068-f005:**
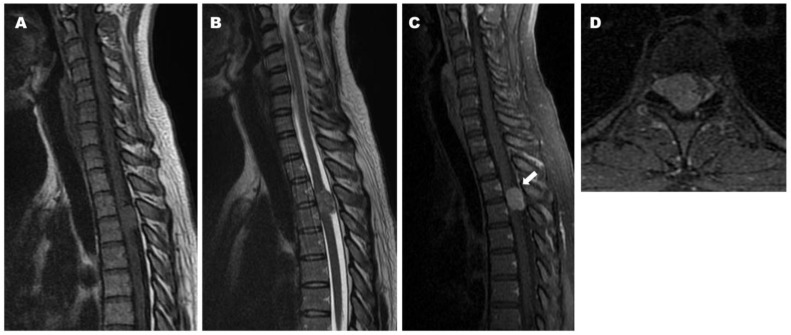
Preoperative magnetic resonance imaging (MRI) demonstrating iso intensity of the tumor and cord compression on both T1- and T2-weighted images ((**A**): Sagittal T1-weighted image, (**B**): Sagittal T2-weighted image). Preoperative sagittal (**C**) and axial (**D**) T1-weighted MRI with contrast demonstrating homogeneous enhancement of the tumor and dural tail sign (arrow).

**Figure 6 jcm-14-06068-f006:**
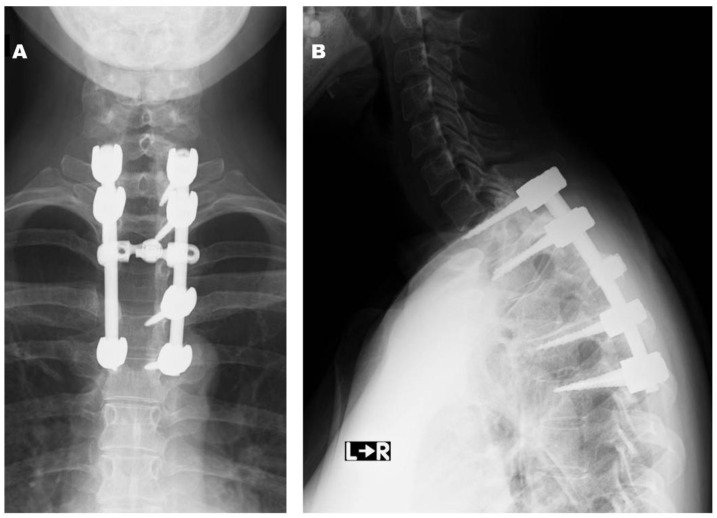
Postoperative radiographs at final follow-up ((**A**): anteroposterior view, (**B**): lateral view).

**Figure 7 jcm-14-06068-f007:**
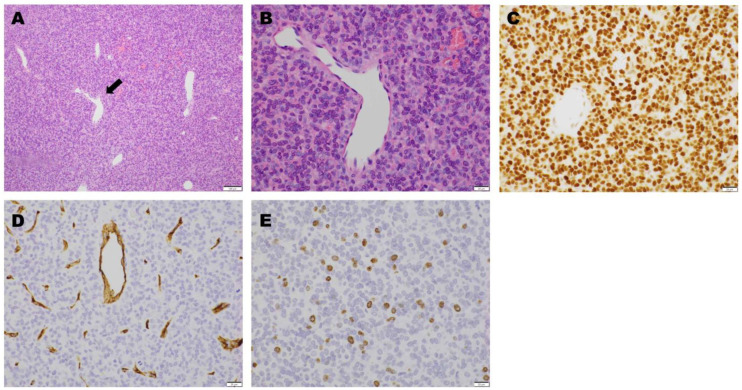
Microscopic findings. Hematoxylin and eosin staining showed sinus-like large vascular branching in a “staghorn” (arrow) pattern ((**A**): magnification, 100×). There was a proliferation of oval- or spindle-shaped tumor cells around the vessels, with slightly acidic cytoplasm and oval nuclei with mitotic figures (>5 mitoses per 10 high-power field). ((**B**): magnification, 400×). On immunohistochemical staining, STAT6 was positive ((**C**): magnification, 400×), while CD34 was negative ((**D**): magnification, 400×). The Ki-67 labeling index was 6–7% ((**E**): magnification, 400×).

**Figure 8 jcm-14-06068-f008:**
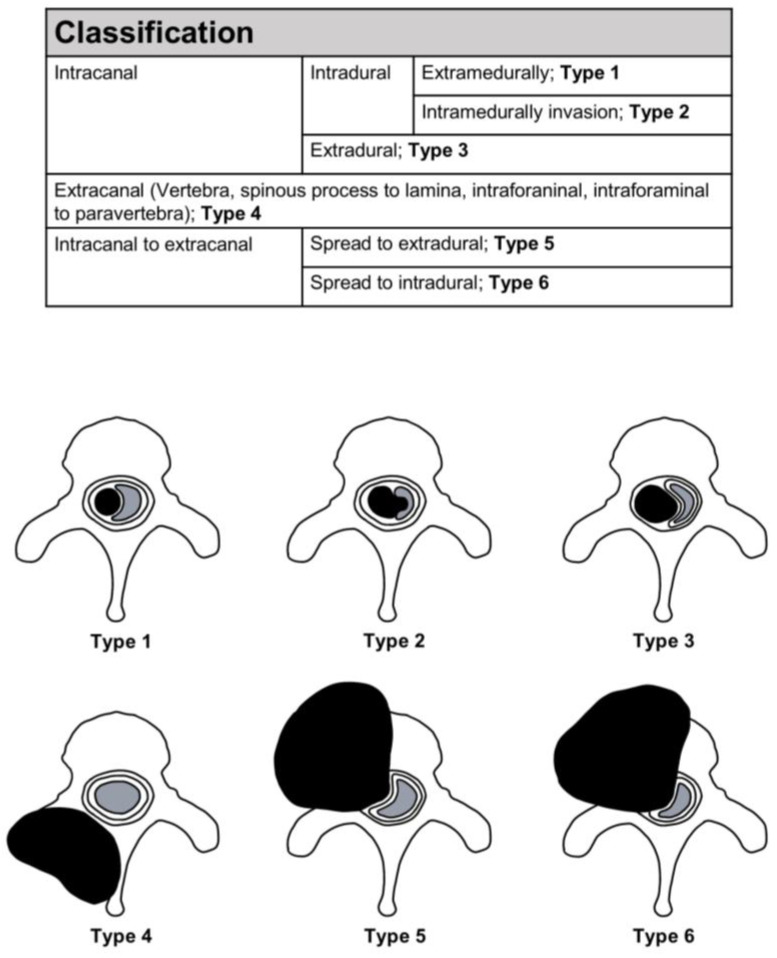
Classification of tumors.

**Table 1 jcm-14-06068-t001:** Literature summary of spinal HPC, WHO grade II or III HPC/SFT, and WHO grade II or III SFT cases.

No.	Authors and Year	Age (yrs)/ Sex	Level	Classification	Treatment	Histology	FU (mos)	Outcome	Notes
**1**	Schirger et al., 1958 [9]	33/F	T1-2	5	GTR + RT	HPC	312	Recurred at 312 mos	
**2**	Kruse et al., 1961 [10]	53/M	C2-3	1	STR + RT	HPC	48	Recurred at 24 mos	
**3**	Pitlyk et al., 1965 [11]	60/M	C4	1	GTR	HPC	NR	NR	
		49/F	C3	1	GTR	HPC	NR	NR	
**4**	Fathie, 1970 [12]	21/M	T6	3	GTR	HPC	NR	NR	
**5**	Gerner et al., 1974 [13]	62/M	L5	NR	RT	N/A	NR	NR	
**6**	McMaster et al., 1975 [14]	NR	T1-2	Extradural type	NR	HPC	NR	NR	
		NR	T6-7	Extradural type	NR	HPC	NR	NR	
		NR	T10	Extradural type	NR	HPC	NR	NR	
		NR	T11	Extradural type	NR	HPC	NR	NR	
		NR	S1	Extradural type	NR	HPC	NR	NR	
**7**	Harris et al., 1978 [15]	28/M	C2-6	5	STR + RT	HPC	55	Disease-free	
		65/F	L2	5	PR + RT	HPC	43	Disease-free	
**8**	Stern et al., 1980 [17]	31/F	C6	5	GTR	HPC	12	Disease-free	
**9**	Cappabianca et al., 1981 [16]	52/F	C6	5	STR + RT	HPC	1	NR	
**10**	Muraszko et al., 1982 [18]	30/M	L3-4	6	STR + RT	HPC	145	Recurred at 129 mos, pulmonary mets at 146 mos	Extremely vascular tumor
		41/F	T11-L3	5	GTR	HPC	NR	NR	Extremely vascular tumor; preoperative embolization
		15/M	T11-L1	5	GTR	HPC	NR	NR	Extremely vascular tumor
		11/F	T10	5	STR + RT	HPC	45	Recurred at 38 mos	Extremely vascular tumor
**11**	Wold et al., 1982 [19]	38/M	Vertbra	NR	Surgery	Benign HPC	70	Disease-free	Four of the five patients died from recurrence or metastases of original HPC
		47/F	Secrum	NR	Surgery	Malignant HPC	17	NR	
		62/F	Secrum	NR	NR	Malignant HPC	NR	NR	
		33/F	Vertbra	NR	Surgery	Malignant HPC	372	Local recurrence at 312 mos	
		37/F	Sacrum	NR	Surgery	Malignant HPC	60	NR	
**12**	Ciappetta et al., 1985 [20]	48/M	C4	2	GTR	HPC	NR	Recurred at 70 mos	
		36/F	C6	5	PR + RT	HPC	24	NR	Required PR because of heavy tumor bleeding
**13**	Yagishita et al., 1985 [21]	NR/F	C7	6	Surgery	HPC	NR	NR	Extremely vascular tumor
**14**	Bridges et al., 1988 [22]	25/F	S1-3	5	STR + RT	HPC	9	NR	Hemorrhagic tumor
**15**	Tang et al., 1988 [23]	19/M	L2	4	Surgery + RT + Chemo	HPC grade II to III	48	Local recurrence at 36 mos	
**16**	Mena et al., 1991 [4]	47/M	T3-4	5	GTR	HPC	96	Disease-free	Extremely vascular tumor
**17**	Salvati et al., 1991 [24]	29/F	L1-3	5	GTR + RT	HPC	120	Disease-free	Extremely vascular tumor
**18**	Cizmeli et al., 1992 [25]	20/M	L2	Extradural type	GTR + RT	HPC	NR	NR	Extremely vascular tumor; preoperative embolization
**19**	Lin et al., 1996 [26]	16/F	C2	4	GTR	HPC	17	Disease-free	
**20**	Dufour et al., 2001 [27]	45/M	Cervical	1	GTR	HPC	24	Disease-free	
		18/F	Dorsal	1	STR + RT	HPC	54	Disease-free	
		43/F	Dorsal	1	GTR	HPC	49	Disease-free	
		38/M	Dorsal	1	GTR + RT	HPC	250	Disease-free	
**21**	Akhaddar et al., 2002 [28]	39/M	T4-6	3	GTR + RT	HPC	36	Disease-free	
**22**	Betchen et al., 2002 [29]	31/M	L4	1	GTR	HPC	6	NR	
**23**	Ijiri et al., 2002 [6]	39/F	L1-2	3	GTR	HPC	24	Disease-free	
**24**	Musacchio et al., 2003 [30]	56/F	Occiput-C1	5	PR	HPC	72	Local recurrence at 24 mos	Preoperative embolization
**25**	Mohammadianpanah et al., 2004 [32]	21/M	T2	5	STR + RT	HPC	NR	NR	
**26**	Kashiwazaki et al., 2007 [33]	31/M	T4-6	2	GTR	HPC	36	Disease-free	
**27**	Kumar et al., 2007 [67]	16/F	T4-5	5	STR + RT	HPC	9	NR	
**28**	Zhao & Zhao, 2007 [34]	NR	C:10 patients T:9 patients L:3 patients S:1 patient	Intradural type: 4 patients; extradural type: 19 patients	GTR, GTR + RT, STR + RT	HPC	NR	One patient had local recurrence at 24 mos	
**29**	Endo et al., 2008 [35]	62/M	L1	2	GTR	HPC	NR	Disease-free	
**30**	Chou et al., 2009 [37]	80/M	T10	2	GTR	HPC	36	Disease-free	
**31**	Fitzpatrick et al., 2009 [36]	54/M	L4-5	4	GTR + RT	HPC	NR	Disease-free	Preoperative embolization
**32**	Zentar et al., 2009 [39]	42/F	S1-2	4	GTR + RT	HPC	5	NR	
**33**	Verbeke et al., 2010 [40]	21/F	S	NR	STR	HPC	60	Disease-free	
		40/M	S	NR	GTR	HPC	60	Lung and bone mets at 36 mos	
		50/M	T12	NR	STR	HPC	nonakasalvati4	Local recurrence at 180, 204, 228, and 240 mos, lung, liver, and soft tissue mets at 252 mos	Patients died from recurrence or metastases of original HPC
		31/F	L4	NR	GTR	HPC	48	Disease-free	
		44/M	S	NR	RT + Chemo	HPC	84	NR	Patients died from original HPC
		55/F	S	NR	RT	HPC	132	Disease-free	
**34**	Ackerman et al., 2011 [41]	58/M	T10	2	GTR	HPC	NR	NR	
**35**	Moscovici et al., 2011 [42]	20/M	T9-10	1	GTR	HPC grade III	24	Disease-free	
**36**	Drazin et al., 2013 [45]	56/ M	Occiput-C4	1	GTR	HPC borderline between grade II and III.	60	Disease-free	There was a small tumor at the T2-3 level. The patient underwent successful radiosurgery of the area.
**37**	Liu et al., 2013 [46]	31/M	L3-4	1	GTR + RT	HPC grade II	32	Disease-free	
		28/M	L3-S2	1	GTR + RT	HPC grade II	36	Disease-free	
		51/M	C2-3	1	GTR	HPC grade III	72	Local recurrence at 26 mos	
		2/M	L1-5	1	STR	HPC grade III	7	Local recurrence at 4 mos	
		44/F	L1-2	1	STR + RT	HPC grade II	252	Local recurrence at 84 mos	
		53/M	C2-3	1	STR + RT	HPC grade II	56	Local recurrence at 24 mos	
		31/F	T5-7	1	STR + RT (radiosurgery) + Chemo	HPC grade II	300	Local recurrence at 72 mos	
		23/F	T12-L1	1	GTR + RT	HPC grade III	72	Local recurrence at 28 mos	
		32/F	T5-6	2	GTR + RT	HPC grade II	79	Disease-free	
		24/M	C5-7	2	GTR + RT	HPC grade II	20	Disease-free	
		25/F	T12-L1	2	STR + RT	HPC grade III	32	Local recurrence at 6 mos	
		31/F	C2-7	5	STR + RT	HPC grade III	18	Local recurrence at 6 mos	
		53/M	T3-5	6	GTR + RT (radiosurgery)	HPC grade II	152	Local recurrence at 53 mos	
		18/M	T11-L2	5	GTR + RT	HPC grade III	64	Local recurrence at 5 mos, pulmonary mets	
		56/F	C2-4	6	STR + RT	HPC grade II	128	Local recurrence at 60 mos	
		14/M	L2	6	STR + RT	HPC grade III	48	Local recurrence at 24 mos, posterior cranial fossa mets	
		25/F	C2-3	3	GTR + RT	HPC grade III	54	Local recurrence at 4 mos	
		37/M	C5-6	3	GTR + RT	HPC grade II	72	Local recurrence at 36 mos	
		46/F	C1-4	5	STR + RT	HPC grade II	25	Disease-free	
		73/F	T9-10	6	STR + RT	HPC grade II	264	Local recurrence at 240 mos	
		44/M	L5-S1	6	STR + RT	HPC grade III	53	Local recurrence at 48 mos	
		19/F	T5-6	3	GTR + RT	HPC grade II	29	Disease-free	
		19/M	C7-T1	3	GTR + RT	HPC grade III	53	Local recurrence at 24 mos, temporal lobe mets	
		14/M	L1-2	5	STR + RT	HPC grade III	24	Local recurrence at 6 mos	
		36/F	C1-2	3	GTR + RT	HPC grade II	120	Local recurrence at 54 mos	
		50/M	T11-12	6	GTR + RT	HPC grade II	65	Disease-free	
**38**	Shirzadi et al., 2013 [47]	56/M	Occiput-C3	1	GTR	HPC grade III	24	Metastatic recurrence at 6 mos	Metastatic recurrence at the T2-3 level. Required radiosurgery for metastatic lesion.
		27/F	T7-8	2	GTR	HPC grade II	36	Disease-free	
		57/M	T9-10	6	GTR + RT	HPC grade II	24	Disease-free	Preoperative embolization
**39**	Zhang et al., 2014 [49]	43/M	C6-T2	5	GTR	HPC grade III	12	Disease-free	Preoperative embolization
**40**	Das et al., 2015 [50]	50/M	C4-5	1	GTR + RT + Chemo	HPC grade III	24	Disease-free	
		12/M	T11-L1	1	GTR + RT	HPC grade III	9	Disease-free	
		34/M	T8-10	2	STR + RT + Chemo	HPC grade III	24	Local recurrence at 24 mos	Deceased from extensive spread of the disease.
		37/F	T7-9	6	GTR	HPC grade II	6	Disease-free	
**41**	Turk et al., 2015 [51]	19/F	C1-2	2	STR	HPC	NR	Disease-free	
		15/F	T9-10	2	GTR	HPC	NR	Disease-free	
**42**	Chew et al., 2017 [52]	63/M	T9	1	STR	HPC grade II	12	Disease-free	
**43**	Yi et al., 2017 [53]	65/M	T12	4	GTR	SFT/HPC grade II	69	Disease-free	
		54/M	T11	4	GTR	SFT/HPC grade II	10	Local recurrence	
		31/M	L4-5	5	GTR	SFT/HPC grade III	51	Disease-free	
		27/M	C2-4	4	GTR	SFT/HPC grade II	13	Local recurrence at 1 mos	
		64/M	T4-S1	Spinal Canal type	GTR	SFT/HPC grade III	NR	NR	
		42/F	T2-3	Spinal Canal type	GTR	SFT/HPC grade II	36	Disease-free	
**44**	Jia et al., 2018 [54]	55/M	T2-3	5	GTR + RT	SFT/HPC grade II	93	Disease-free	
		36/F	C7-T1	5	GTR	SFT/HPC grade II	75	Local recurrence at 73 mos	Preoperative embolization
		45/M	T5-6	5	GTR	SFT/HPC grade II	77	Local recurrence at 70 mos	Preoperative embolization
		36/F	T7	4	GTR	SFT/HPC grade III	33	Disease-free	Preoperative embolization
		49/F	L3	5	GTR + RT	SFT/HPC grade III	37	Local recurrence at 23 mos	Preoperative embolization
		57/M	L1	5	STR + RT	SFT/HPC grade II	73	Local recurrence at 53 mos, lung and ribs mets	
		57/F	T7-9	4	GTR + RT	SFT/HPC grade II	41	Disease-free	Preoperative embolization
		42/M	T8	4	GTR + RT	SFT/HPC grade III	46	Local recurrence at 37 mos	
		46/F	C5-6	5	GTR + RT	SFT/HPC grade II	13	Disease-free	
		33/M	C4-5	5	STR + RT	SFT/HPC grade II	7	Disease-free	
		23/F	C5-6	4	STR + RT	SFT/HPC grade III	30	Local recurrence at 16 mos	
		42/F	C2	5	GTR + RT	SFT/HPC grade III	24	Disease-free	Preoperative embolization
		49/F	T9	5	STR + RT	SFT/HPC grade III	24	Local recurrence at 17 mos, lung and brain mets	
		55/M	T1	5	GTR + RT	SFT/HPC grade III	20	Disease-free	Preoperative embolization
		57/M	L5-S2	5	STR	SFT/HPC grade III	18	Local recurrence at 12 mos	Preoperative embolization
		35/F	T4	5	GTR	SFT/HPC grade II	34	Disease-free	
**45**	Shukla et al., 2018 [56]	13/M	T8	1	GTR	Anaplastic HPC grade III	48	Disease-free	
**46**	Wang et al., 2018 [57]	65/M	L2	1	GTR + RT	HPC grade II	48	Disease-free	
**47**	Fujita et al., 2019 [58]	50/M	T7	1	GTR	SFT/HPC grade II	12	Disease-free	
**48**	Li et al., 2019 [59]	35/F	T6-7	5	GTR	HPC	12	Disease-free	
**49**	Louis et al., 2019 [60]	82/ NR	T9-10	1	STR + RT	HPC grade III	5	NR	
**50**	Paeng, 2019 [61]	57/M	T5-6	2	GTR	HPC	24	Disease-free	
**51**	Fiorenza et al., 2020 [62]	73/F	T5-7	5	GTR + RT	SFT/HPC grade II	36	Disease-free	
**52**	Singla et al., 2020 [63]	50/M	C4-5	1	GTR + RT + Chemo	SFT/HPC grade II	62	Local recurrence at 12 mos	
		37/F	T7-9	6	GTR	SFT/HPC grade II	53	Disease-free	
		12/F	T12-L1	1	GTR + RT + Chemo	SFT/HPC grade III	52	Disease-free	
**53**	Nishii et al., 2021 [64]	68/F	O-C3	1	STR	SFT/HPC grade II	18	Disease-free	
**54**	Olmsted et al., 2021 [65]	81/F	T11-12	1	STR	SFT grade III	6	Disease-free	
**55**	Okubo et al., 2023 [66]	68/M	Cervical	1	PR	SFT/HPC grade III	118	Local recurrence at 34 mos	Required PR because of the amount of bleeding during surgery obscures the relevant surgical field or intraoperative spinal cord monitoring indicates deterioration in the motor evoked potential
		65/F	Lumbar	2	PR	SFT/HPC grade II	38	Local recurrence at 5 mos	
		55/M	Thorax	1	PR	SFT/HPC grade III	50	Disease-free	
		49/M	Thorax	1	PR	SFT/HPC grade III	58	Local recurrence at 35 mos	
	Present cases	40/F	T12-L1	3	GTR	SFT grade II	228	Local recurrence at 72 mos	
		36/F	T3-4	1	GTR	SFT grade III	36	Disease-free	

SFT: solitary fibrous tumor; HPC: hemangiopericytoma; FU: follow-up; yrs: years; mos: months; M: male; F: female; GTR: gross total resection; RT: radiotherapy; STR: subtotal resection; PR: partial resection; NR: not reported; mets: metastasis; Chemo: chemotherapy.

**Table 2 jcm-14-06068-t002:** Demographic and tumor characteristics.

Variable	Value
**Age (yrs)**	
Mean (range)	40.8 (2–82)
Not reported	29 (18.5%)
**Sex**	
Male	68 (43.3%)
Female	60 (38.2%)
Not reported	29 (18.5%)
**Level**	
Occipitocervical	4 (2.6%)
Cervical	39 (24.8%)
Cervicothoracic	3 (1.9%)
Thoracic	63 (40.1%)
Thoracolumbar	8 (5.1%)
Lumbar	24 (15.3%)
Lumbosacral	3 (1.9%)
Sacral	11 (7.0%)
Not reported	2 (1.3%)
**Compartment (Classification)**	
Type 1. Intracanal–intradural–extramedurally	34 (21.7%)
Type 2. Intracanal–intradural–intramedurally invasion	14 (8.9%)
Type 3. Intracanal–extradural	9 (5.7%)
Type 4. Extracanal	11 (7.0%)
Type 5. Extradural to extracanal	33 (21.0%)
Type 6. Intradural to extracanal	12 (7.7%)
Details are unknown	44 (28.0%)

## Data Availability

The authors confirm that the data supporting the findings of this study are available within the article.

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
