# Peer review of "WHO Grade II or III Solitary Fibrous Tumors (Hemangiopericytomas) of the Spine: Two Case Reports with a Comprehensive Review of the Literature"

_jcm, 2025, doi:10.3390/jcm14176068_

Round 1
Reviewer 1 Report
Comments and Suggestions for Authors
I was invited to review a paper titled “Solitary fibrous tumors WHO grade II or III (Hemangiopericytomas) of the spine: two case reports with a comprehensive review of the literature” for the Journal of Clinical Medicine. This manuscript presents two illustrative cases of spinal WHO grade II and III solitary fibrous tumors (SFTs), alongside a comprehensive review of 157 cases from the literature. The authors have made a commendable effort in compiling a thorough clinical, pathological, and statistical overview of this rare entity. The case presentations are well-documented, and the review is structured clearly, offering valuable insights into the management, recurrence patterns, and prognostic factors of spinal SFTs. Please find my comments below:
- Literature Review Methodology: While databases are listed (PubMed, Scopus, Embase), a more explicit description of the search strategy, inclusion/exclusion criteria, and PRISMA flow (even in a supplementary figure) would enhance reproducibility.
- Terminology Consistency: Use of “HPC” throughout may be confusing, given the WHO 2021 reclassification. The authors should consistently use “SFT” with historical context clarified at the outset.
- Statistical Limitations: While the multivariate analysis is useful, the large proportion of missing data (especially WHO grades) should be more critically acknowledged as a limitation of the study.
- Treatment Recommendations: The discussion appropriately highlights GTR as favorable, but the conclusion could better emphasize the nuanced decision-making required in surgically complex or high-risk cases where radiosurgery may be preferred.
- Language and Grammar: A thorough language edit is recommended to correct occasional grammatical issues and improve overall fluency.
Author Response
Comments 1: Literature Review Methodology: While databases are listed (PubMed, Scopus, Embase), a more explicit description of the search strategy, inclusion/exclusion criteria, and PRISMA flow (even in a supplementary figure) would enhance reproducibility.
Response 1: The identification of studies via databases was added as s Supplementary Figure, and “The inclusion criteria were that the primary tumor was in the spine, and that reports included not only surgical cases but also cases treated with radiotherapy (RT), chemotherapy, and no treatment. Exclusion criteria included reports of distant metastasis to the spine from other sites and recurrence after tumor removal (those without detailed descriptions of the initial treatment).” was added to main text (P7, L152-157).
Comments 2: Terminology Consistency: Use of “HPC” throughout may be confusing, given the WHO 2021 reclassification. The authors should consistently use “SFT” with historical context clarified at the outset.
Response 2: Everything except for explanations of terms and tables has been standardized to SFT.
Comments 3: Statistical Limitations: While the multivariate analysis is useful, the large proportion of missing data (especially WHO grades) should be more critically acknowledged as a limitation of the study.
Response 3: At the end of the Discussion section, the following was stated as a research limitation (P22, L357-361): One of the limitations of this study is that the factors used in the multivariate analysis, particularly the WHO grades, contained a large number of missing data, which may have influenced the results. By continuing to increase the number of cases, missing data will be supplemented, and future research will involve creating treatment algorithms based on tumor location and malignancy.
Comments 4: Treatment Recommendations: The discussion appropriately highlights GTR as favorable, but the conclusion could better emphasize the nuanced decision-making required in surgically complex or high-risk cases where radiosurgery may be preferred
Response 4: The goal of surgery remains the same: GTR. The following modification was made accordingly in the Conclusion section (P22, L366-368). “GTR is recommended to treat these lesions. When GTR is difficult, such as in recurrent cases, radiosurgery is one of the most important alternatives for treating spinal SFTs.”
Comments 5: Language and Grammar: A thorough language edit is recommended to correct occasional grammatical issues and improve overall fluency.
Response 5: The English has been corrected by a native speaker.
Reviewer 2 Report
Comments and Suggestions for Authors This study analyzes two SFT cases and reviews clinical data from 157 SFT patients, conducting both univariate and multivariate analyses. While this article offers valuable insights on this rare tumor, some content requires revision.1. The abstract is too simple and does not fully reflect the author's work. For example, the retrospective analysis of the literature is not mentioned. It is suggested to supplement the methods, results and conclusions of the literature research.
2. The abbreviation is mixed. For example, after the first appearance of SFTs, it is sometimes SFT and sometimes SFTs in the following text. It is suggested to unify it. Other abbreviations also need to be carefully checked.
The literature review summarizes patient demographics, pathology, and surgical outcomes, exploring recurrence factors. The introduction lacks this context. Clearly state the literature review's purpose and method to highlight its novelty.
4. The magnification of B/C/D in Figure 2 and Figure 7 is not indicated; Figure 4 and Figure 6 do not indicate when the postoperative images were taken? Are there long-term follow-up images?
5. The 157 patients' clinical information characteristics are summarized in line 140. It is suggested to add a subheading to make this part of the results more clear. The data in Table 2 is not clearly labeled, such as adding Sex, N(%), Level, N(%), and modify the other rows accordingly; the Age column needs to be carefully considered in terms of presentation form.
6. Is Figure 8 self-drawn or a figure from a reference? It is not marked in the text.
7. The content from line 162 to 212 is suggested to be presented in a table.
8. The content of 4. Comparison between the recurrence or metastasis cases and disease-free cases should also belong to the content of the literature review. Can it be considered as a secondary title of 3. Review of the Literature? For example: 3.1 Literature Search, 3.2 Summary of General Information and Surgery of Patients, 3.3 Comparison between the recurrence or metastasis cases and disease-free cases.
Author Response
Comments 1: The abstract is too simple and does not fully reflect the author's work. For example, the retrospective analysis of the literature is not mentioned. It is suggested to supplement the methods, results and conclusions of the literature research.
Response 1: Another reviewer suggested that the rarity of the two cases and the novelty of this study be stated in the Abstract. In line with the comments from reviewer 2, we have added the following (P1, L19-34): This report describes two cases of WHO grade II and III SFTs in the spine and presents a review of the literature. In the first case, an extradural WHO grade II SFT recurred 6 years after the first surgery, and a second surgery was performed, including wide excision of the surrounding tissue. The patient has remained recurrence-free for 16 years since the second surgery. In the second case, an intradural extramedullary WHO grade III SFT was resected, including the dura mater, and the patient has remained recurrence-free for 3 years since the surgery. Few reports have described tumor recurrence and long-term outcomes after reoperation, as in the first case, or extensive resection including the dura, as in the second case. Furthermore, the literature review not only summarizes patients’ general and surgical information, but also indicates, based on multivariate analysis, that gross total resection (GTR) is an important factor in preventing recurrence and metastasis. This is the first study to comprehensively examine previous reports and identify risk factors for recurrence and metastasis. In addition, because recurrences have been reported long after surgery, we believe that even if GTR is performed surgically, it is important to conduct follow-ups to check for long-term recurrence.
Comments 2: The abbreviation is mixed. For example, after the first appearance of SFTs, it is sometimes SFT and sometimes SFTs in the following text. It is suggested to unify it. Other abbreviations also need to be carefully checked
Response 2: Terminology has been standardized as much as possible within the context.
Comments 3: The literature review summarizes patient demographics, pathology, and surgical outcomes, exploring recurrence factors. The introduction lacks this context. Clearly state the literature review's purpose and method to highlight its novelty.
Response 3: We have added the following to the Introduction section (P2, L61-64): Furthermore, the literature review summarizes patient demographics, pathology, and surgical outcomes, exploring factors contributing to recurrence. This is the first study to comprehensively examine previous reports and describe risk factors for recurrence and metastasis.
Comments 4: The magnification of B/C/D in Figure 2 and Figure 7 is not indicated; Figure 4 and Figure 6 do not indicate when the postoperative images were taken? Are there long-term follow-up images?
Response 4: We have added that the magnification was 400 and that the X-p was taken at the final follow-up.
Comments 5: The 157 patients' clinical information characteristics are summarized in line 140. It is suggested to add a subheading to make this part of the results more clear. The data in Table 2 is not clearly labeled, such as adding Sex, N(%), Level, N(%), and modify the other rows accordingly; the Age column needs to be carefully considered in terms of presentation form.
Response 5: The subheading was added as fellows: Age and Sex, Spinal Level and Compartment, MRI Appearance and Differential Diagnosis, Treatment, Tumor Origin, Histopathology, Clinical Outcome.
Comments 6: Is Figure 8 self-drawn or a figure from a reference? It is not marked in the text
Response 6: This is an original drawing that I did myself.
Comments 7: The content from line 162 to 212 is suggested to be presented in a table.
Response 7: These are summarized in Table 3, and we have added (Table 3) where appropriate (P17, L214 and P18, L230).
Comments 8: The content of 4. Comparison between the recurrence or metastasis cases and disease-free cases should also belong to the content of the literature review. Can it be considered as a secondary title of 3. Review of the Literature? For example: 3.1 Literature Search, 3.2 Summary of General Information and Surgery of Patients, 3.3 Comparison between the recurrence or metastasis cases and disease-free cases.
Response 8: As we have pointed out, we have summarized it with three subtitles.
Reviewer 3 Report
Comments and Suggestions for Authors
This is a significant contribution, given the rarity of this entity, particularly in its spinal location. The detailed clinical, radiologic, surgical, and pathological documentation is commendable. I have the following suggestions that may improve clarity, strengthen the scientific impact, and enhance the manuscript’s reproducibility. I would like to commend you on presenting these two cases of WHO grade II and III solitary fibrous tumors (SFTs) of the spine.
1. Abstract: Please revise the abstract to align with a concise case report format and to highlight the novelty of your work. The revision should place greater emphasis on: 1) The rarity of WHO grade II and III solitary fibrous tumors in the spine. The inclusion of both a recurrent extradural lesion and an intradural extramedullary lesion. Finally, the long-term follow-up outcomes after gross total resection (GTR).
In the conclusion sentence, explicitly state the key clinical takeaway from these cases and the accompanying literature review. You may condense or relocate background details (such as the historical case count) so that these novel aspects remain the primary focus.
2. Introduction
-
Shorten background information on SFT/HPC nomenclature changes to focus more on the gap in current literature regarding spinal SFTs.
-
Clearly highlight the novelty of your report: two high-grade spinal SFTs with different anatomical locations and management outcomes, including long-term follow-up.
-
Specify why these cases are of particular interest in relation to recurrence, tumor location, and surgical strategy.
3. Literature Review & Discussion
-
While the literature review is comprehensive, it could be streamlined to emphasize: 1) Differences in recurrence rates, survival, and metastasis between intracranial and intraspinal SFTs. 2) The role and timing of adjuvant radiotherapy or radiosurgery for residual or recurrent lesions.Comparative outcomes of GTR versus subtotal resection, particularly in the spinal setting. 3) Discuss the relevance of the long-term follow-up in Case 1 (13 years) and how it supports recommendations for extended surveillance. In addition, consider adding a short paragraph summarizing proposed treatment algorithms based on tumor location, grade, and patient comorbidity.
This is a rare and clinically valuable case report, and I commend you for your thorough documentation and analysis. Addressing the points outlined in my detailed comments will help improve the clarity, focus, and educational value of your manuscript, ensuring it has the greatest possible impact for readers and clinicians.
Author Response
Comments 1: Abstract. Please revise the abstract to align with a concise case report format and to highlight the novelty of your work. The revision should place greater emphasis on: 1) The rarity of WHO grade II and III solitary fibrous tumors in the spine. The inclusion of both a recurrent extradural lesion and an intradural extramedullary lesion. Finally, the long-term follow-up outcomes after gross total resection (GTR). In the conclusion sentence, explicitly state the key clinical takeaway from these cases and the accompanying literature review. You may condense or relocate background details (such as the historical case count) so that these novel aspects remain the primary focus.
Response 1: The article has been revised to emphasize the novelty of each case and to state the importance of long-term follow-up. In addition, in Case 2, tumor resection, including dura mater, was one of the novelties. The following has been added to the discussion (P21, L313-316): As in Case 2, when the tumor is tightly adhered to the dura mater in a location where it is relatively easy to remove, such as within the spinal canal, the possibility that the dura mater is the tumor origin cannot be ruled out, so removal of the dura mater as well should be considered.
Comments 2: Introduction. Shorten background information on SFT/HPC nomenclature changes to focus more on the gap in current literature regarding spinal SFTs.
Clearly highlight the novelty of your report: two high-grade spinal SFTs with different anatomical locations and management outcomes, including long-term follow-up.
Specify why these cases are of particular interest in relation to recurrence, tumor location, and surgical strategy.
Response 2: It was difficult to shorten the background information on SFT/HPC nomenclature change any further, as shortening it would have made it unclear whether multiple terms were searched for in this review. In addition, regarding the gaps in current literature on spinal SFTs, we have added the following (P2, L51-54): Although these tumors are understood to share a biologic basis, they remain behaviorally distinct; WHO grade I SFTs occurring in the CNS tend to be classified as benign[3], while WHO grade II or II are aggressive, known to metastasize systemically and commonly recur[4–6].
The novelty of the case was described as follows (P2, L59-61): There are few reports of extradural tumor recurrence and long-term outcomes after reoperation, and intradural extramedullary tumors that have been resected, including the dura mater.
Comments 3: Literature Review & Discussion. While the literature review is comprehensive, it could be streamlined to emphasize: 1) Differences in recurrence rates, survival, and metastasis between intracranial and intraspinal SFTs. 2) The role and timing of adjuvant radiotherapy or radiosurgery for residual or recurrent lesions. Comparative outcomes of GTR versus subtotal resection, particularly in the spinal setting. 3) Discuss the relevance of the long-term follow-up in Case 1 (13 years) and how it supports recommendations for extended surveillance. In addition, consider adding a short paragraph summarizing proposed treatment algorithms based on tumor location, grade, and patient comorbidity.
Response 3: Following suggestions from other reviewers, we have made the changes to make it easier to understand by adding headings rather than simply writing a long, drawn-out description. Also, since it is difficult to comment on the treatment algorithm at this time, I have listed it as a topic for future research.
“One of the limitations of this study is that the factors used in the multivariate analysis, particularly the WHO grades, contained a large number of missing data, which may have influenced the results. By continuing to increase the number of cases, missing data will be supplemented, and future research will involve creating treatment algorithms based on tumor location and malignancy. (P22, L357-361)”
Reviewer 4 Report
Comments and Suggestions for Authors
The manuscript includes the long term outcome in two cases of SFT, while including also a short review regarding the topic. The literature includes according references, however very few mentions of long term outcomes can be found according to my knowledge, thus the manuscript may be of interest to the readers.
The contemporary approach is to devide the review from the case report. However, authors present a case based review providing also an appropriate statistical analysis, providing a more comprehensive understanding of the topic with a more regorous analysis and stronger evidence.
please consider changing the following:
Lines 38-40 please rephrase and provide reference
Line 48 - the patient was treated 20 years prior to the study, state the exact follow-up in months or years.
lines 74-79 - could you provide also microscopic findings of the second operation?
Table 1 is confusing - maybe try downsizing letters, redesigning the table with different layout so less pages are used, with less page use.
Figure 8. If possible improve quality
References seem appropriate.
Thank you
Author Response
Comments 1: Lines 38-40 please rephrase and provide reference
Response 1: Added literature.
Comments 2: Line 48 - the patient was treated 20 years prior to the study, state the exact follow-up in months or years.
Response 2: We have added that it was 22 years ago.
Comments 3: lines 74-79 - could you provide also microscopic findings of the second operation?
Response 3: This is pathology from surgery carried out 16 years ago, and we will not be able to provide the pathology photos in time for the deadline.
Comments 4: Table 1 is confusing - maybe try downsizing letters, redesigning the table with different layout so less pages are used, with less page use.
Response 4: The font size has been reduced.
Comments 5: Figure 8. If possible improve quality
Response 5: We are currently doing this.
Round 2
Reviewer 2 Report
Comments and Suggestions for Authors
The author made revisions according to my suggestions, but there are still several details that need improvement.
-The author mentioned 'Based on the morphological data in the literature, spinal SFTs were divided into six types (Figure 8)', but relevant references were not cited here. Please add related references.
-Line 256: The title number should be 3.3 Comparison between Recurrence or Metastasis Cases and Disease-free Cases.
-Since statistical analysis was conducted in this study, it is recommended to include information about the statistical software (name, version, …).
Author Response
Comments 1: The author mentioned 'Based on the morphological data in the literature, spinal SFTs were divided into six types (Figure 8)', but relevant references were not cited here. Please add related references.
Response 1: Since this is based on information from many studies listed in Table 1, we have added them as references.
“Based on the morphological data in the literature[6, 9–30, 32–37, 39–42, 45–47, 49–54, 56–67], spinal SFTs were divided into six types (Figure 8):”
Comments 2: Line 256: The title number should be 3.3 Comparison between Recurrence or Metastasis Cases and Disease-free Cases.
Response 2: We have corrected it that way.
Comments 3: Since statistical analysis was conducted in this study, it is recommended to include information about the statistical software (name, version, …).
Response 3: The statistical analysis has been added as follows: Statistical analyses were conducted using JMP Pro 17 Software (SAS Inc., Cary, NC, USA) for Windows.
Reviewer 4 Report
Comments and Suggestions for Authors
I would like to thank the authors for addressing all of my questions and concerns.
I understand the difficulty of providing further histopathology images from a case treated such a long time ago.
The manuscript is adequately written and the long follow up of patients is scarce in the literature and will certainly be of interest to some readers, thus I believe that the manuscript can be accepted with some further improvements.
Please check the tables
regarding table 1 I would suggest changing the No with No of patients, in any other case the reader may be confused, expecting that this is the number of studies, or adjust the numbers so that they mach the numbers of studies which may even be a better solution. I believe that the table will be better presented in that way.
Also please adjust the tables in a way that they look more tidy, eg table 3 factor(s), but the s is one line below, it should be factors all in one line or radiotherapy should star with a capital letter as all the other words start with one. Please tidy the tables so they look better if published. The optic quality of the manuscript is always associated with its' quality by the readers.
I would suggest not including your 2 cases in the flow chart in the supplementary figure.
The histopathological images provide scales that are not easily recognized however and are not uniform. Also the quality of the figures is not as high as I would expect.
The English language use is adequate, however can be improved in some parts, please read once more the text.
Thank you!
Author Response
Comments 1: regarding table 1 I would suggest changing the No with No of patients, in any other case the reader may be confused, expecting that this is the number of studies, or adjust the numbers so that they much the numbers of studies which may even be a better solution. I believe that the table will be better presented in that way.
Response 1: The counted number has been changed to the number of studies instead of the number of patients.
Comments 2: Also please adjust the tables in a way that they look more tidy, eg table 3 factor(s), but the s is one line below, it should be factors all in one line or radiotherapy should star with a capital letter as all the other words start with one. Please tidy the tables so they look better if published. The optic quality of the manuscript is always associated with its' quality by the readers.
Response 2: The font size has been reduced to fit as much as possible into a single column, and the first letter of radiotherapy has been changed to uppercase.
Comments 3: I would suggest not including your 2 cases in the flow chart in the supplementary figure.
Response 3: Our two cases have been excluded from the flow chart.
Comments 4: The histopathological images provide scales that are not easily recognized however and are not uniform. Also the quality of the figures is not as high as I would expect.
Response 4: All scales have been unified. Also, image quality has been improved as much as possible.
Comments 5: The English language use is adequate, however can be improved in some parts, please read once more the text. We have attached a certificate of English editing.
Response 5: The English has been already been edited twice by a native English speaker, but it may be difficult to do it any more than this.
Additionally, Figure 8, which was requested to be revised last time, has now been revised.